# Measurable health effects associated with the daylight saving time shift

**Hanxin Zhang**[1,2], **Torsten Dahlén**[3], **Atif Khan**[2], **Gustaf Edgren**[3,4], **Andrey Rzhetsky**[1,2,5]*

**1** Committee on Genetics, Genomics and Systems Biology, The University of Chicago, Chicago, Illinois, United States of America, **2** Department of Medicine, and Institute of Genomics and Systems Biology, The University of Chicago, Chicago, Illinois, United States of America, **3** Department of Medicine Solna, Clinical Epidemiology Division, Karolinska Institutet, Stockholm, Sweden, **4** Department of Cardiology, Södersjukhuset Hospital, Stockholm, Sweden, **5** Department of Human Genetics and Committee on Quantitative Methods in Social, Behavioral, and Health Sciences, The University of Chicago, Chicago, Illinois, United States of America

* andrey.rzhetsky@uchicago.edu

**Data Availability Statement:** All data and code underlying our study and necessary to reproduce our results are available on Github: https://github.com/hanxinzhang/dst/tree/master/data.

## Abstract

The transition to daylight saving time (DST) is beneficial for energy conservation but at the same time it has been reported to increase the risk of cerebrovascular and cardiovascular problems. Here, we evaluate the effect of the DST shift on a whole spectrum of diseases—an analysis we hope will be helpful in weighing the risks and benefits of DST shifts. Our study relied on a population-based, cross-sectional analysis of the IBM Watson Health MarketScan insurance claim dataset, which incorporates over 150 million unique patients in the US, and the Swedish national inpatient register, which incorporates more than nine million unique Swedes. For hundreds of sex- and age-specific diseases, we assessed effects of the DST shifts forward and backward by one hour in spring and autumn by comparing the observed and expected diagnosis rates after DST shift exposure. We found four prominent, elevated risk clusters, including cardiovascular diseases (such as heart attacks), injuries, mental and behavioral disorders, and immune-related diseases such as noninfective enteritis and colitis to be significantly associated with DST shifts in the United States and Sweden. While the majority of disease risk elevations are modest (a few percent), a considerable number of diseases exhibit an approximately ten percent relative risk increase. We estimate that each spring DST shift is associated with negative health effects–with 150,000 incidences in the US, and 880,000 globally. We also identify for the first time a collection of diseases with relative risks that appear to decrease immediately after the spring DST shift, enriched with infections and immune system-related maladies. These diseases' decreasing relative risks might be driven by the documented boosting effect of a short-term stress (such as that experienced around the spring DST shift) on the immune system.

**Funding:** This work was funded by the DARPA Big Mechanism program under ARO contract W911NF1410333; by National Institutes of Health grants R01HL122712, 1P50MH094267, and U01HL108634-01; and by a gift from Liz and Kent Dauten. The funders had no role in study design, data collection and analysis, decision to publish, or preparation of the manuscript.

**Competing interests:** The authors have declared that no competing interests exist.

## Author summary

Over a quarter of the world population is subjected to the daylight saving time (DST) shift twice a year, which disrupts both human work and rest schedules and possibly the body's biological clock. Several clinical studies have reported an increased risk of cerebrovascular and cardiovascular problems with DST shifts but little is known about other potential health effects. The DST shift represents a natural exposure experiment which allows us the unique opportunity of linking health outcomes to an external, state-wide event in the US and Sweden. We performed a comprehensive, phenome-wide screening of the putative health effects of the DST shift by analyzing two independent, country-scale health data-sets, and found both adverse and protective associations with DST shifts in several clusters of conditions. We successfully verified previously reported associations, such as heart diseases and injuries, and identified new signals–for example, immune-related conditions. We suggest that the ramifications of daylight-saving time shifts should be acknowledged and further tested.

## Introduction

The idea of introducing daylight saving time (DST) was attractive at the time of candles and gas lamps, as it allowed workers to use sunlight a bit longer during working hours as well as saving employers' energy for lighting. Much has changed since then. Today, only a small fraction of electricity expenditure actually corresponds to producing light after sunset (in the US, it is about six percent in the residential sector and eight percent in the industrial sector) [1]. Yet, over a quarter of the world population is subjected to the DST shift twice a year, which disrupts both human work and rest schedules and possibly their circadian clock rhythms [2]. DST shifts have been shown to have a measurable effect on electric power consumption, although not necessarily in the intended direction [3,4]. Previous studies have demonstrated that the spring DST shift causes noticeable alterations in human behavior in terms of waking-up time and self-reported alertness, [5] a significant increase in fatal traffic accidents (up to 30 percent on the day of commencing DST), [6] a short-term rise in workplace injuries (5.7 percent after the spring DST shift as employees sleep 40 minutes less on average), [7] and elevated rates of acute myocardial infarction (up by about 3.9 percent) [8]. The study described in [6] and [9] reported conflicting results regarding whether the DST shifts are associated with accident incidence. The study described in [10] found increased mental health- and behavioral health-oriented emergency department visits in certain seasons, but did not obtain conclusive results on whether they could be linked to the DST shifts.

Remarkable progress has been made in the past decade towards understanding the neurology of sleep-wake cycles and circadian rhythms, and how they affect our behavior [11]. Despite these advances, significant gaps remain in our knowledge of how changes in the social clock (DST shifts) interact with the body's biological clock and impact human health. Recent studies have urged for investigations into the clinical implications of DST shifts on human health [12,13].

The DST shift represents a natural exposure experiment which allows us the unique opportunity of linking health outcomes to an external, state-wide event in the US and Sweden. Earlier analyses of DST shift effects typically examined a single medical condition per study, often with conflicting or inconclusive results [6,9,10]. In addition, these studies often relied on small, disease-specific datasets with thousands of observations from a single country or a single hospital, making it impossible to run phenome-wide screening. In the present study, we used

the electronic health records (EHRs) of hundreds of millions of people across two countries, for the purpose of: (1) examining the temporal disease risk dynamics in relationship to DST shifts, and; (2) identifying those population strata which manifest health changes linked to DST-related schedule disruption.

## Methods

### Data and Other Materials

Our study accessed EHRs from two countries: In the US, through the IBM Watson Health MarketScan dataset, [14] and in Sweden, through the Swedish national inpatient register [15]. The version of the MarketScan dataset we used in this study incorporated health information about more than 150 million unique patients, observed during the time interval between 2003 and mid-2014. Individuals followed asynchronous enrollment and disenrollment on insurance policies, leading to variance in their "visibility" durations and endpoints in the data. The mean follow-up time for the patient in the US MarketScan database was 154 weeks. The Swedish register described more than nine million unique Swedes, nearly all observed continuously from 1968 to 2011, except in cases of death or emigration.

In both datasets, disease diagnoses were represented with codes defined by the World Health Organization (WHO) International Classification of Diseases (ICD) taxonomies, versions 9 and 10 for the US, and versions 8, 9, and 10 for Sweden, along with a day-level temporal label recording the date of diagnosis in the US data or the discharge date in the Swedish data. To evaluate the risk of DST shifts across the whole spectrum of diseases, we grouped ICD codes into 263 condition classes under 31 biological systems using the WHO ICD-10 guidelines (S1 Table) [16]. The grouping is hierarchical and exhaustive, so neighboring codes fall in the same or similar condition classes, and no ICD code is left uncategorized.

### Statistical Analyses

Taking advantage of this ICD code grouping, we summarized the daily incidences of each class of conditions for females and males in the following age groups: 0–20 years old (or, alternatively, in the larger US dataset, separately 0–10, and 11–20), 21–40, 41–60, and over 60. Building on a previous study's methodology [8], we quantified the relative risk (RR) involved with shifting to and from DST by comparing the diagnosis rate for each day during the week following a DST shift to the linear expectation, which is the average diagnosis rate of the same week day, two weeks before and two weeks after the day of interest (please refer to S1 Appendix Fig A for additional clarification). The diagnosis rate is the expected proportion of patients diagnosed with a given disease on the day of interest out of all enrollees at a given time point. We estimated week-level RR values by collapsing incidences reported during the whole week following a DST shift and comparing it to the average of corresponding week-level diagnoses rates two weeks before and after the time point. We corrected the possible effect of holidays (see the S1 Appendix Section 1.5 for a full list of holidays considered) and different day lengths (23 hours at the spring DST shift and 25 hours at the autumn). We obtained all the RR estimates in a Bayesian framework and shrunk them towards one. The shrinkage represented our prior belief that we expected the DST shifts to show no health effects for most conditions. The Bayesian procedure pooled information about multiple diseases–with a hierarchical prior distribution imposed over RR estimates. This shrinkage technique resolved the multiple comparisons problem and also yielded statistically efficient estimates [17]. To provide our readers with a frame of reference, we supplemented all of our Bayesian analyses with their frequentist counterparts (see S1 Appendix Section 1.6). The results of our Bayesian and frequentist analyses were very similar after a false coverage rate (FCR) adjustment of frequentist confidence

intervals (see S1 Appendix Section 1.7), [18] suggesting a practical equivalence of FCR-corrected frequentist and hierarchical Bayesian (with shrinkage prior) frameworks.

Bayesian and frequentist methods should produce compatible but not identical results. Each method comes with its own advantages and disadvantages. The most obvious difference between them is associated with specification of a prior distribution over parameter values. In our Bayesian analysis, we assumed that most of the estimated RR values would be close to one (shrinkage assumption, implemented as a prior distribution strongly pulling estimates towards the central mean). This assumption forces the results to be conservative (which eliminates weaker signals) and removes the need to correct the results for multiple tests. The frequentist analysis is agnostic in regard to likely distribution of priors. Some statisticians argue that this type of analysis is less subjective and easier to interpret. The frequentist analyses require an explicit correction for multiple statistical tests and are likely to produce estimates with larger absolute deviations from one.

To control for possible false positive discoveries, we designed a few negative control experiments. Because DST was not adopted in Sweden until 1980, we compared the RRs of time transitions before 1980 (at some "pseudo-DST" shift points, *i.e.*, when time shift *would* have happened, see S1 Appendix Section 1.8 for details) and after 1980 at real DST shift points. For the US data, we analyzed all patients residing in states not observing DST as a negative control. Furthermore, we introduced another negative control by repeating the RR estimation procedure at "pseudo-DST" shift dates, which were set to 28 days *after* each real DST shift in the spring and 28 days *before* each real DST shift in the autumn. The latter negative control resulted in the most statistically powerful test among the three, because it covered the largest population comparable in size to the groups being tested for association.

Because we ran all our analyses in parallel in both Bayesian and frequentist frameworks, we decided to present Bayesian results in the main text, highlighting the differences between the two approaches when relevant. The decision to analyze inpatient data separately was driven by the consideration that patients who were hospitalized ("inpatient") may have been subject to fewer of the social and environmental confounders that drive spurious associations. Inpatient admissions are typically associated with a set of health problems distinct from outpatient visits, with more severe conditions, such as acute heart attacks, most commonly treated in hospital inpatient settings.

## Results

In the US inpatient cohort, we detected a significant risk elevation in a number of disease and condition groups (see S1 Appendix 2.1 and the summary table); for example, complications related to pregnancy, childbirth, and puerperium (PCP), as well as injuries, symptoms, and signs across various systems, and circulatory diseases (Fig 1A). We observed stand-out increases in the RR for some injuries, immune disorders, heart diseases, and possibly in related conditions such as renal failure (urinary system-related, shown in S2 Table but not Fig 1A as it is not among the top 30 for the effect size) and circulatory/cognition symptoms and signs (Fig 1A). Signals of relative risk change presented in Fig 1 were automatically selected according to their effect size and for their significance as shown in a comparison between the experimental and negative control tests (see S1 Appendix Section 2.1). For the whole spectrum of conditions considered in this study, we present the RR changes with DST shifts in S2 Table. The results of experiments with the negative controls are shown in S3 Table. We conducted similar analyses with the frequentist approach and found results consistent with those using the Bayesian framework. In some cases, we even noticed larger estimated effect sizes after adjusting for FCR (S4 Table and S5 Table).

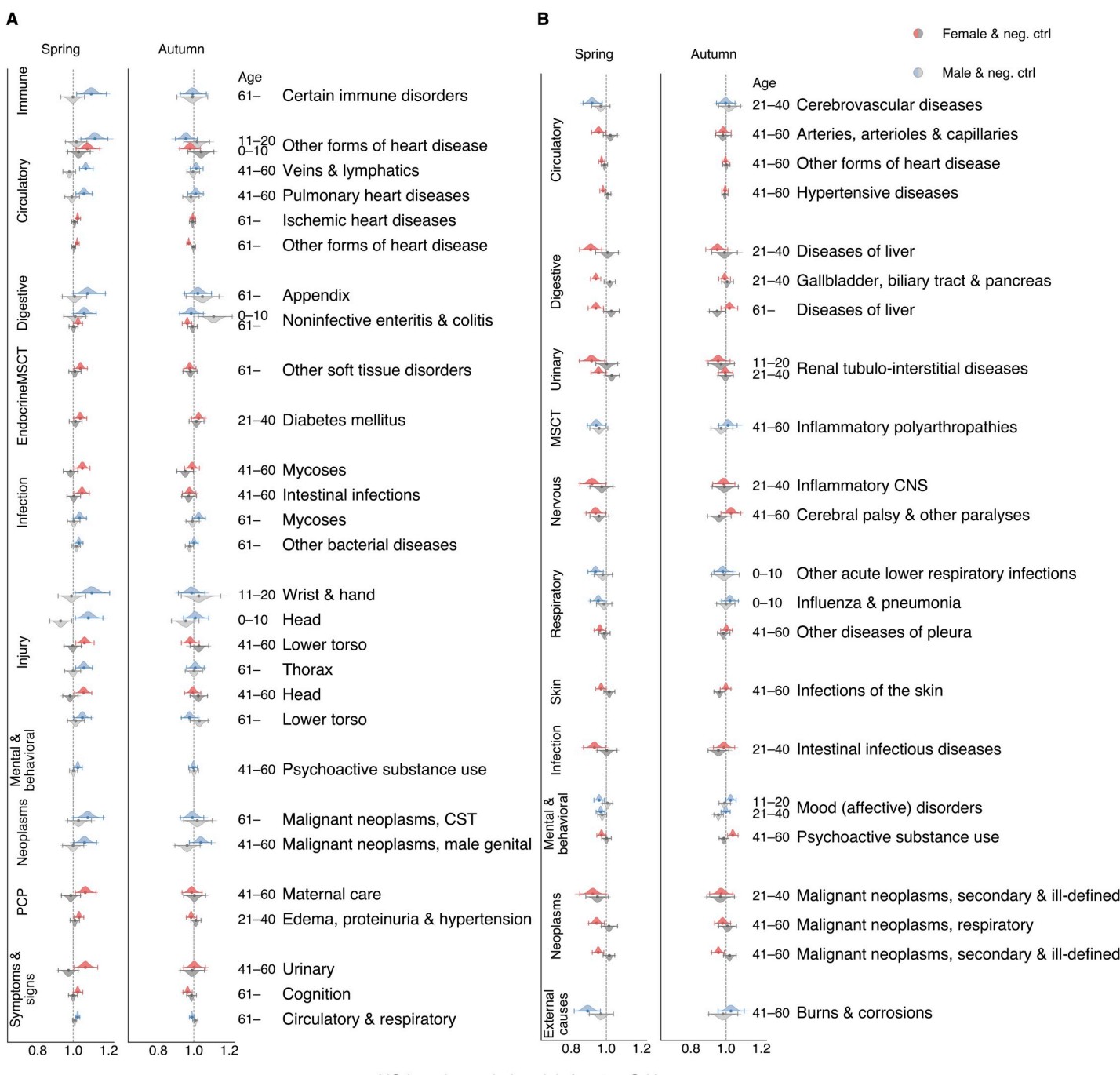

US inpatient relative risk (99.9% C.I.)

**Fig 1. Daylight saving time (DST) shifts appear to affect the relative risk (RR) of numerous diseases spanning several human biological systems.** Color-coded violin plots and error bars represent RR estimates' posterior density distributions and credible interval (CI) boundaries, respectively. We adjusted the credible intervals (CI) for multiple tests with a Bayesian shrinkage procedure that ensured 99.9 and 99 percent significance levels for US and Swedish data, respectively. Gray-colored violin plots and error bars indicate analogous RR distribution results computed for the negative control (pseudo-DST shift dates, the same populations as for the real DST shift dates). The pseudo-DST shift date was selected at 28 days after the real DST shift in spring and 28 days before the real one in autumn. For the Swedish tests, negative controls were performed on data before 1980 when DST was not observed in Sweden. We selected all depicted signals automatically (see S1 Appendix Section 2.1), as the significant RR largest in effect size that were: (1) significantly greater than one in the spring DST shift analyses (colored), and: (2) *not* significantly greater than one in the negative control (gray) analyses, or *vice versa* for decreased signals. We also excluded too broadly defined, ambiguous clinical and laboratory findings, examinations, and health services from the figures (they are retained in the Supplementary Tables). **(A)** The top 30 conditions exhibiting the largest increasing RRs (effect sizes) for the results of the US inpatient analyses. The results suggest risk expansion in diseases involving the immune, circulatory, and digestive systems, the musculoskeletal system and connective tissue (MSCT), the endocrine systems, some infections and injuries, mental and behavioral disorders, neoplasms, problems with

pregnancy, childbirth, and the puerperium (PCP), and symptoms and signs across various systems. **(B)** All disease conditions with significantly decreased RR in inpatient data after the spring DST shift (minus ambiguous procedures).

To the best of our knowledge, we are the first to report the DST-related RRs of disorders involving the digestive system (such as noninfective enteritis and colitis), which rose three percent after the spring DST shift in females over 60 and six percent in males under ten.

We also observed, for the first time, that the RR for a number of diseases appears to reduce after the spring DST shift (see Fig 1B). Such diseases include a set of infectious and inflammatory diseases. In the Bayesian analysis, the effect sizes of the negative RR changes tend to be smaller than those for positive RR changes. The gray contour in Fig 2A shows the joint distribution estimation of the spring DST's RR versus the negative control's RR in inpatients, spotlighting those increased and decreased RR change signals (see S1 Appendix Section 2.1 for more details).

In the smaller Swedish dataset, as expected, we found fewer significant RR change signals. Under a less conservative significance level (99 percent versus 99.9 percent in the US analyses), we were able to reproduce the RR change signals for a subset of cardiovascular diseases in the elderly population (Fig 2B, S6 Table, and S7 Table). The RRs of some heart and cerebrovascular diseases go up in the spring, when the day length shrinks by one hour, but not in the autumn. As would be expected for a real effect, the RR for circulatory diseases increased after 1980 in the spring DST shift but not in the autumn one. Corresponding frequentist results are shown in S8 Table and S9 Table. Interestingly, the RR of psychoactive substance use increases as much as nine percent with the spring DST shift and 12 percent with the autumn DST shift but only among males age 20 or above population (Fig 2B and S6 Table and S7 Table).

## Discussion

Our analyses reproduced the major past literature's findings, such as an elevation in ischemic heart disease rates in males and females older than 60, [8,19–21] and a rise in accidents [6,22] and injuries [7]. We also discovered novel significant RR change signals, such as a DST-shift-associated increase in mental and behavioral disorders due to the aforementioned elevated psychoactive substance use in the male adult population. The strongest effect size was observed among males between the ages 41–60 and the signal was consistent in both the US and Sweden. A large body of studies have shown that circadian disruption increases the risk of substance abuse [23–26], with some studies providing in-depth mechanistic details [27]. Because psychoactive substance users generally have very disrupted diurnal rhythms [28], it seems plausible that further acute disruption due to DST shifts may lead to abnormal clock function, resulting in increased vulnerability for substance abuse.

The findings in the US "all-patients" dataset (S1 Appendix Fig B, S10–S13 Tables) resolves the inconclusive results of a previous study focusing only on emergency admissions [10]. To the best of our knowledge, never reported before, immune-related disorders tend to become more common than expected in the first week following each spring DST shift. The largest effect sizes is observed for the following conditions: an approximate ten percent increase in the RR for some cardiovascular and heart diseases in inpatients under 20, injuries at various locations and ages (in the frequentist framework, RR estimates increased by 30 percent), and some immune disorders in senior males. The absolute risk posed by the DST shift is discussed in Section 2.5 of the S1 Appendix. The comparison of Bayesian and frequentist analyses indicates that there is not much data in support of the estimates computed for a subset of diseases, and that the prior has a strong influence on the estimate.

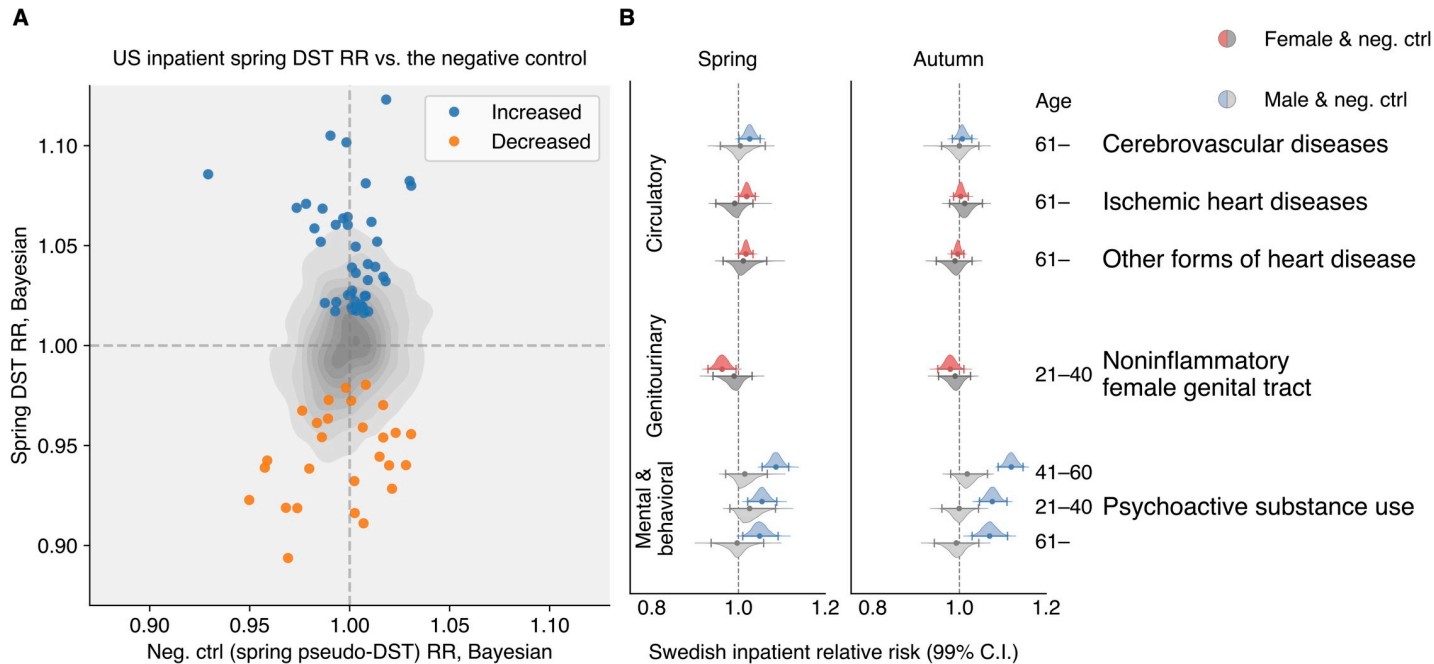

**Fig 2.** **(A)** Spring RR estimates versus the negative control results in the US inpatient population. The gray contour represents the empirical estimation of the joint distribution for all RR estimates. The blue and orange markers accent the increased and decreased signals selected by an impartial procedure based on effect size and significance (see S1 Appendix Section 2.1) **(B)** All conditions showing significant change around the DST shift analyses in the Swedish data after 1980. None of their corresponding negative controls were significantly different from one. As in the US data, we observed an increased RR in ischemic and other forms of heart disease in the senior population and mental and behavioral disorders due to psychoactive substance use in middle-age males. The RR for cerebrovascular diseases in senior inpatients increased in Sweden in the week following the spring DST shift, confirming the increase (with no statistical significance) in US inpatients. By contrast, in the US *all-patient* population, cerebrovascular diseases actually decreased significantly (S10 Table and S12 Table).

Our analysis identified several population strata that appeared responsive to the time and schedule disruption in terms of changes in disease RR: (1) very young (0–10, or 11–20) and older (41–60 and over 60) patients with (probably preexisting) chronic diseases, for example, acute myocardial infarction, behavioral and emotional disorders, and stress-related immune disorders such as inflammatory bowel diseases and noninfective enteritis; (2) children (0–10) and teenagers/young adults (11–20) who were prone to accidents resulting in injuries of the head, wrist, and hand; and (3) older adults (41–60 and older than 60) who were more likely to injure the lower torso or thorax. Note that while MarketScan represents the United States fairly evenly, geographically, it excludes the uninsured population.

In a DST shift effect analysis, one should keep in mind that there are two distinct phenomena in play: (1) the natural variation of day length, the amount of light over the year, and the daily sunlight intensity cycle affects the human circadian clock, behavior, and numerous diseases [29–45], and; (2) the disruption of individuals' daily schedule due to the DST shift. Our analyses were designed to target the effects associated with the latter, adjusting for the former with negative controls. Furthermore, it is not always possible to distinguish between increased diseased incidence (for a truly new diagnosis) and increased care (for an established ongoing diagnosis). For a subset of patients under observation in the US dataset since their birth, we were able to carefully identify the first disease diagnosis and validate the RR change signal for noninfective enteritis and colitis after the spring DST shift (see S1 Appendix Sections 1.10 and 2.4). We performed an analogous "first-diagnosis" analysis for older US patients in the all-patients dataset, which resulted in only one statistically significant result: an increased rate of "other forms of heart disease" in females over 60 (see S1 Appendix Sections 1.10 and 2.4).

We have considered using a more complicated model, such as a mixed-effect regression model, which would allow us to include some chronic diseases as confounding factors to obtain better estimates of disease relative risks. We did not attempt to control for any particular disease for two reasons: (1) introducing control diseases would cause a combinatorial explosion in the model space, and; (2) controlling for specific diseases would exponentially increase the number of necessary statistical tests. Both issues would increase analysis complexity, make interpretation more difficult, and decrease our ability to detect RR change signals. Therefore, we decided against using this approach.

What mechanism could plausibly explain the reduced RR observed for some of the infectious and immune diseases during the spring DST shift? The spring DST shift might act as a short-term stressor, as one hour is subtracted from a normal night's schedule. A transient, mild stress was shown to enhance the immune system (as opposed to long-term stress, which has the opposite, suppressive action), possibly accounting for the reduced RR for urinary, skin, and other infections. As F.S. Dhabhar put it, a "short-term stress can enhance the acquisition and/or expression of immune-protective (wound healing, vaccination, anti-infectious agent, anti-tumor) or immuno-pathological (pro-inflammatory, autoimmune) responses." [44,45] Of course, this is but one possible explanation and possible alternative explanations could be suggested.

## Limitations

This study suggests that even a one-hour change of the clock may impact population health significantly, but a number of caveats accompany this assertion.

First, it is important to keep in mind that diseases are not truly independent of each other; one illness can facilitate the development of another, and environmental insults may also lead to the exacerbation of a (pre-existing) chronic disease. Therefore, our analysis cannot distinguish between "driver" and "passenger" diseases. Note that the low $p$-values and narrow credible/confidence intervals are driven by the large data set. This does not necessarily mean that the data set is free of bias, and any bias will propagate and can very well drive associations.

Second, while the US dataset records the actual diagnoses dates (rather than "billing dates"), chronic diseases, such as hypertension and diabetes, are likely to have been developing for a long time before the actual diagnoses were made and recorded. It is possible that environmental insult (mild stress) acts as a trigger to worsen an already pre-existing conditions, so that patients are forced to see a physician (which results in diagnoses entered in records). The Swedish national registry records the "discharge dates" instead of the actual dates of diagnosis. In most cases, neither the date of diagnosis as in the US data nor the discharge date is expected to be the same as the actual time of disease attack. But these dates still more accurately reflect the time of attack than the "billing dates" do. We tried to smooth out the discrepancy between the dates by adding up all codes of interest in the following week of the daylight saving time change, and estimated the week-level RR based on it.

Third, it is possible that disease coding errors could influence our RR estimates. A simple way to spot miscoding is to look for sex-specific codes assigned to the wrong genders. For instance, the data shows males having pregnancy, ovarian cancer, and females having prostate cancer. There are two scenarios to explain the origins of such errors: Either the sex was recorded wrongly or the code itself is inaccurate. The former scenario would not affect our analyses substantially because we anticipated for symmetric coding errors in both sexes that offset against each other. On the other hand, if a diagnosis itself is miscoded, it may influence the RR estimation and tests. We estimated that the miscoding rate at the MarketScan data is around 0.52 percent, (see S1 Appendix Section 1.12. Data Limitations, for details of this

estimation). The error rate is positive but is small in comparison to the observed DST shift effect sizes. Importantly, simple disease coding errors are unlikely to be in any way related to DST shifts, and would only bias RR estimates towards the null model.

Fourth, the major difficulties in our analysis were associated with changing coding standards (especially in Sweden, which went from ICD8 to ICD9, then to ICD10) and insufficient data sample despite the fact that datasets spanned whole countries. In particular, we did not have sufficient data to analyze every condition in every age/sex group. We alleviated the coding problem using two synergistic measures: (1) careful ICD version mapping, and; (2) considering relatively large constellations of diseases instead of following very specific conditions (for example, we analyzed "ischemic heart diseases," including acute myocardial infarction, instead of specifically "acute myocardial infarction"). We felt that these categories of larger disease groups were robust enough to withstand changes in medical practice and diagnosis criteria. The larger disease group we used in our analysis also helped with increasing disease-specific patient sample size, as the collective incidence of a group of diseases is the sum of incidences of individual diseases in the group.

## Supporting information

**S1 Appendix. Complete details of materials and methods.**
(PDF)

**S1 Table. A mapping from ICD-10 codes to conditions and diseases.**
(CSV)

**S2 Table. Week-level RR estimates of the US inpatient analysis, via the Bayesian method.**
(CSV)

**S3 Table. Week-level RR estimates of the US inpatient analysis on pseudo-DST shift dates as a negative control, via the Bayesian method.**
(CSV)

**S4 Table. Week-level RR estimates of the US inpatient analysis, via the frequentist method.**
(CSV)

**S5 Table. Week-level RR estimates of the US inpatient analysis on pseudo-DST shift dates as a negative control, via the frequentist method.**
(CSV)

**S6 Table. Week-level RR estimates of the Swedish inpatient analysis since 1980, via the Bayesian method.**
(CSV)

**S7 Table. Week-level RR estimates of the Swedish inpatient analysis before 1980 as a negative control, via the Bayesian method.**
(CSV)

**S8 Table. Week-level RR estimates of the Swedish inpatient analysis since 1980, via the frequentist method.**
(CSV)

**S9 Table. Week-level RR estimates of the Swedish inpatient analysis before 1980 as a negative control, via the frequentist method.**
(CSV)

**S10 Table. Week-level RR estimates of the US all-patient analysis, via the Bayesian method.**
(CSV)

**S11 Table. Week-level RR estimates of the US all-patient analysis on pseudo-DST shift dates as a negative control, via the Bayesian method.**
(CSV)

**S12 Table. Week-level RR estimates of the US all-patient analysis, via the frequentist method.**
(CSV)

**S13 Table. Week-level RR estimates of the US all-patient analysis on pseudo-DST shift dates as a negative control, via the frequentist method.**
(CSV)

**S14 Table. A mapping from ICD-9-CM to ICD-8.**
(CSV)

**S15 Table. A mapping from the US modification of ICD-8, 9, 10 to conditions.**
(CSV)

**S16 Table. A mapping from the Swedish modification of ICD-8, 9, 10 to conditions.**
(CSV)

**S17 Table. A count summary of some female-specific diseases in the US data set.**
(CSV)

**S18 Table. A count summary of some male-specific diseases in the US data set.**
(CSV)

**S19 Table. A summary of conditions with increased RRs (Bayesian, US all-patient).**
(CSV)

**S20 Table. A summary of conditions with increased RRs (frequentist, US all-patient).**
(CSV)

**S21 Table. A summary of conditions with increased RRs (Bayesian, US inpatient).**
(CSV)

**S22 Table. A summary of conditions with increased RRs (frequentist, US inpatient).**
(CSV)

**S23 Table. A summary of conditions with decreased RRs (Bayesian, US all-patient).**
(CSV)

**S24 Table. A summary of conditions with decreased RRs (frequentist, US all-patient).**
(CSV)

**S25 Table. A summary of conditions with decreased RRs (Bayesian, US inpatient).**
(CSV)

**S26 Table. A summary of conditions with decreased RRs (frequentist, US inpatient).**
(CSV)

**S27 Table. Estimation of cost associated with the DST shift (Bayesian, US all-patient).**
(CSV)

**S28 Table. Estimation of cost associated with the DST shift (frequentist, US all-patient).** (CSV)

**S29 Table. Estimation of cost associated with the DST shift (Bayesian, US inpatient).** (CSV)

**S30 Table. Estimation of cost associated with the DST shift (Frequentist, US inpatient).** (CSV)

## Acknowledgments

We are grateful to E. Gannon, R. Melamed, and M. Rzhetsky, for comments on earlier versions of this manuscript.

## Author Contributions

**Conceptualization:** Hanxin Zhang, Gustaf Edgren, Andrey Rzhetsky.

**Data curation:** Hanxin Zhang, Torsten Dahlén, Atif Khan, Gustaf Edgren.

**Formal analysis:** Hanxin Zhang, Torsten Dahlén, Atif Khan.

**Funding acquisition:** Gustaf Edgren, Andrey Rzhetsky.

**Investigation:** Hanxin Zhang, Atif Khan, Gustaf Edgren, Andrey Rzhetsky.

**Methodology:** Hanxin Zhang, Atif Khan, Gustaf Edgren, Andrey Rzhetsky.

**Project administration:** Andrey Rzhetsky.

**Resources:** Gustaf Edgren, Andrey Rzhetsky.

**Software:** Hanxin Zhang, Andrey Rzhetsky.

**Supervision:** Gustaf Edgren, Andrey Rzhetsky.

**Validation:** Torsten Dahlén, Atif Khan.

**Visualization:** Hanxin Zhang.

**Writing – original draft:** Hanxin Zhang, Gustaf Edgren, Andrey Rzhetsky.

**Writing – review & editing:** Hanxin Zhang, Torsten Dahlén, Atif Khan, Gustaf Edgren, Andrey Rzhetsky.

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
