## [Decision Letter · Decision Letter 0]

12 Feb 2020

Dear Dr. Rzhetsky,

Thank you very much for submitting your manuscript "Measurable Health Effects Associated with the Daylight Saving Time Shift" for consideration at PLOS Computational Biology.

As with all papers reviewed by the journal, your manuscript was reviewed by members of the editorial board and by several independent reviewers. In light of the reviews (below this email), we would like to invite the resubmission of a revised version that takes into account the reviewers' comments.

The reviewers were overall quite positive about the manuscript, but made many concrete requests for clarifications. Two of the reviewers also recommended correcting for additional potential confounding factors, which should be possible to do given the data used.

We cannot make any decision about publication until we have seen the revised manuscript and your response to the reviewers' comments. Your revised manuscript is also likely to be sent to reviewers for further evaluation.

Sincerely,

Lars Juhl Jensen

Associate Editor

PLOS Computational Biology

Virginia Pitzer

Deputy Editor

PLOS Computational Biology

Reviewer's Responses to Questions

**Comments to the Authors:**

Reviewer #1: The paper presents a large-scale analysis of how daylight-saving time (DST) affects people’s health for a wide range of diseases using EMR data. They use US market scan with around 120 million patients and the Swedish national patient registry with 9 million patients. They use Bayesian inference to estimate the effect size and include a frequentist approach for reference. They reproduce diseases earlier shown to be affected by DST and come with novel results, like pregnancy related diagnoses and mental diseases. They provide literature references to substantiate these and show that they reproduce in both the US and Swedish data.

The paper presents original research. Although DST time shift have been researched before, no one have done it this systematically for as many different diseases as here. The statistical method seems to be a standard Bayesian approach without any novelties in it. With a minor exception it seems rigorous and sound. The main novelties are that they include a wide range of diseases in one study and use a larger population than used in earlier studies of DST effect on health. For example, they state that the DST effect on pregnancy diagnoses have never been studied before. The article has relevance both for the scientific community and for decision makers who are arguing over stopping DST. It contains conclusions only possible due to the large data set. It is my impression that the paper should be published with minor revision. However, I have a number of questions, which I would like the authors to address.

Regarding the calculation of relative risk: The sentence about which days are compared to which (week after DST to +/- two weeks) can be complicated to understand, and I suggest putting in a figure reference to fig S2.

As the authors mention as a weakness, there might be other causes that drive the difference in disease rate than DST. However, the authors have chosen not to control for other than age, and gender (and race/ethnicity?). Given the data there it is possible to control for pre-existing diseases/diagnoses in the patients, even in large-scale analysis. Or to repeat a selection of the significant results in a mores strictly controlled manner. I would like the authors comments on why they did not choose to include control for any other diseases.

A very crucial detail in the author’s work is that they made negative controls. In this way, they demonstrate that it is the DST having an effect on the results and that it is not reproducible in a random week. However, I find it highly questionable to use data from prior to 1980 as negative controls in Sweden. These numbers represent a different era of medicine than the newest data and coding practices might have changed significantly. I see their validity to demonstrate that the trend starts with introduction of DST, but not as comparison to the general numbers.

The data contains coding in both ICD versions 8, 9, and 10. There are very big differences going from version 9 to 10. Have the authors investigated if the coding frequency change abruptly because of the change? And have they verified in any way that this does not affect their results.

The author’s mention inconclusive results from reference 6, 12, and 13 (p 5, l 60). What are the results? They later write that these the inconclusive results from reference 13 are resolved (p. 10 l 179-180), but does not clearly state which results were resolved and how.

Did the authors have any way of distinguishing between acute and planned/follow-up visits? It seems likely that the acute visits are more affected by the DST than visits planned from before. E.g. for the injuries mentioned p. 10 l 193-194, where planned visits are probably not related to the DST. Could this change the result of the analysis?

The results show a change of childbirth RR (p 8 l 146). If the number of childbirth changes, so does the number of complications related to childbirths. Could this not alone explain the other maternal codes (p 8 l 144)? I.e. does the rate of complications increase more than the observed increase in birthrate?

In page 10 l 90 the authors mention that chronic diseases were probably preexisting. How was this investigated? There is no documentation for this statement, which should be easy to produce (i.e. number of cases where the chronic disease was seen before). Was such analysis carried out for other diseases?

In supplementary section 2.4 (p 15), the significant first-time signal is dismissed as “likely to be due to seasonal convexity or concavity”. I assume that the numbers are calculated with the same +/- 14 days comparison and that this handles the seasonal convexity or concavity. How come that the main analysis does not have the same problem?

It is not clear what the goal of the geographic location comparison is, if it can a priori be dismissed because many other covariates could influence the signal.

The authors present three sets of results with Bayesian, “half Bayesian” and frequentist methods. All the comparison is in the supplementary methods. Can they provide more information on the advantages / disadvantages for these methods in the main section?

I unfortunately do not know a lot about Bayesian statistics, and therefore I am not able to assess the choice of models. In the interest of the uninformed reader, I would ask that the authors explain their method in a way more readable to people with less knowledge about Bayesian statistics. I give examples of what I would like to be clearer here.

It is not clear at all how the actual data comes in to the model. In frequentist statistics, you estimate the parameters by maximizing the likelihood function. The authors give a formula to calculate the diagnosis rate on the time point of interest from the RR (5). But is the RR not the value you want to estimate and p0 a value you can calculate from the data? It would help understanding if the authors made more clear what loss function is being minimized / maximized to estimates the parameters.

In the interest of understanding the flow of the model, I don't understand how the x* (6a-6c) affects the model. They are not mentioned in the formulas for estimating the RR, but depends on the p* values.

The authors correct for the number of enrolled patients in the US data. As far as I understand they compare data points within +/- 2 week from the DST shift. Is the change in enrollment really so large that you have to correct for difference over a few weeks? Could they give a number on how much it changes on average? Or is it to correct for the yearly difference? In the latter case, it can be argued that the Swedish population also has changed during the study period.

Regarding the Bayesian correction for multiple testing, I don't understand how it works. I would suggest that the authors give more background to explain this. As I understand, the method forces the RR values towards the mean (i.e. 1 or neutral value?) when there is insufficient data. This makes a lot of sense. However, it is not clear how this shrinking is affected by the number of RR values that are estimated. And it is not clear how you can set the different thresholds for the US and Swedish data. Also, I find that the sentence regarding the shrinking in the main method-overview section should be extended to explain.

I assume that all the author’s choices of priors are well thought off and perhaps even standard choices. It would help if they cite some method papers that informs the reader about these choices.

The need to create the corrected RR value is well argued. However, I do not understand how the expected RR ( E[RR] hat ) is calculated and how this solves the problem. Is the expected RR the distribution's mean across all RR estimates that naturally have the temporal change in RR factored in?

Also, there are no comment on choice of parameters for the optimization. Are there any non-standard choices that might be relevant for readers with knowledge about such models?

The typesetting of Supplementary Materials is off: There are several unintended space, some of the letters also almost overlap, and the line spacing is also varying much.

Examples

Suppl Materials page 3, section 1.1 Mappings

All ICD-10 codes are categorized i n this grouping, so i t i s also exhaustive.

Suppl Materials page 4 btw formula (4) and (4a)

"mean and l et i nformation flow [...] comparisons problem i s alleviated

Reviewer #2: The study by Zhang et al presents a phenome-wide study of the effects of daylight savings time and disease incidence. The study is interesting, and it is encouring to see results for both men and women. Some of the results piques my interest, although I would like to see a more thorough dissemination of the results. I was also disappointed to not see a more through comparison of the two cohorts in the main text. Surely the advantage of working with multiple cohorts must be to replicate the findings and/or perform a meta analysis. I also have some concerns with the study design, statistical analysis, and terminology throughout the article. I find it very positive that the authors have provided all code and data in a Github Repo. I also see that the authors actually report some absolute risk differences. I think this would be much more intuitive, compared to the ratios used throughout the article.

INTRODUCTION

The introduction reads nicely, and provides a timely summary of the subject. One minor thing is I believe the appropriate terminology for this kind of analysis is "natural experiment" and not "large-scale intervention experiment". The authors note that previou studies typically focus on a small number of diseases. There is actually a perfectly good reason for this, which I will get back to.

METHODS

The description of the two data sources seem incomplete. Surely not only the Swedish inpatient registry was used. This would not contain information on death/emigration.

L72: "visibility" is typically called "under observation" in epidemiological jargon. Likewise, subjects are "censored" when they are not under observation. It would be very benefecial to provide average/mean follow-up time for each individual.

L72: In all fairness, how are these patients garantued to be unique? Are they linked through a social security number similar to Sweden?

L75-76: unclear why individuals that are dead or or have emigrated are excluded. This is very atypical. Please provide clarification.

From the Data description it is not clear which date is used for the patients. The admission or discharge date?

L85: It is not clear at all how the incidence is calculated. If incidence is calculated as #cases / population at risk, the age ranges are far too wide in my opinion. Also, are cases from the previous year considered in the next year? Or are they completely excluded from the calculation? This can also be seen in the Results section, where women aged 41-60 have an increased incidence of maternal care codes. The vast majority of this group of women will not be at risk to even be pregnant. This goes back to the point as to why previous studies typically deal with a limited number of conditions: defining the population at risk is not trivial, and it is something the authors need to consider in more detail for each condition. Just using the full population will severely deflate and provide nonsense incidence estimates.

L95-98: I would really like to see a correction for year as well.

L98-101: The specification of the Bayesian model is poorly explained. Please provide more detail. Additionally, I am not sure what kind of pooling the authors are using. The main text suggestes a complete pooling, but the Supplementary Material looks like a no pooling scheme to me. The multiple comparison problem is not necessarily solved by just specifying priors. If, however, the authors had used a partial pooling scheme across diagnoses, this would indeed solve the multiple comparison problem. See e.g. the 8 schools example by Gelman and Gelman's Bayesian Data Analysis, 3rd ed page 101. Given that the authors work in the ICD-10 space it should be possible to make use of that heirarchy in a model with partial pooling across diagnoses.

How was the convergence of the Bayesian model assessed? From the github repository, it seems that the authors used PyMC3. PyMC3, to my knowledge, uses the HMC-NUTS that provides a lot of diagnostics. Please also add a citation to PyMC3 in the Main Text.

L100: The use of a conservative prior is good, but the prior specifications in the Supplementary Material need a lot more justification. Also, the use is non-informative priors is generally not recommended. The authors should use priors that put more emphasis on the null value if that is the prior belief.

L113: by "not observing", do the authors mean that these states do not have DST?

L121: when was the decision made to analyse inpatient only? And in which data set?

L124: I disagree with the notion of urgent problems in inpatient care. What I believe the authors mean is that there are a different kind of diagnosis associated with each encounter. E.g. pneumonia, sexually transmitted diseases, etc may be handled in outpatient or GP care. Likewise, routine surgeries represent a fairly low risk and non-acute inpatient group.

RESULTS

L128-129: Could the author write how many, or refer to a table? just refering to "a number" seems very imprecise.

L131: What are stand-out increases? Do the author mean relative large effect sizes?

L134-135: "Signal" - would that be the change in relative risk? What is "automatically detected according to their effect size"?

L140: This is unclear. Did the point estimate of the frequentist confidence interval increase after FCR correction? Or do the authors mean that the frequentist point estimate is greater (in absolute terms), compared to the Bayesian analysis? The latter is expected due to the shrinkage, and should not come as a surprise to the authors.

L142-143: I am not convinced about this result due to the lack of a proper at-risk group. The group at-risk should, in this case, be pregnant women, and not just all women.

L159: Is it to be understood that the only signal that replicates are the cardiovascular diseases under a less stringent treshold? This is very disappointing, and in my opinion this strongly indicates that there is a bias in the data set that confers these increases that is unrelated to DST. A population of 9 million individuals is not a small data set, and it should have plenty of power to detect differences in incidence for single diseases.

In general, it would be good to see the all results simultaneously to see if there are trends within disease areas, using e.g. the ICD-10 chapter structure.

DISCUSSION

L179-180: not clear what exactly is meant here. What is the previous result, and why was it inconclusive?

L184: was this result presented previous? A _ten_ percent increase in cardiovascular and heart disease in a population of children and teenagers seems very extreme. How many children actually had a cardiovascular disease? An absolute risk would give more insight into the actual consequence.

L184-185: I am not a big fan of presenting both Bayesian and Frequest estimates jointly. The Bayesian model that the authors have specified naturally shrinks the estimates towards the null, so it is no surprise that point estimate is smaller than the frequentist procedure. This also indicates, strongly, that there is not a lot of data in support of the values, and that the prior has a strong influence on the estimate.

L189-190: Could these groups very well be the individuals that are covered by an insurance plan? I would imagine children and teenagers are covered by their parents insurance plan.

L195: What does "emergency care" refer to here? Were they admitted to the ER?

L198-199: without any previous studies exploring this, this is pure unfounded speculation and should be stated as such.

L202: This is not garantueed. Insured individuals have a completely different socioeconomic profile versus the uninsured, and extrapolating between the two requires information on the uninsured. This is a general concern when using insurance data, namely that it is highly biased towards specific socioeconomic profiles. This could be explored in the Swedish data set, as there should be registries containing information regarding employment for these dates.

L210-215: I am surprised to see this analysis done in the US data set, and not the Swedish data set, as this would cover more age groups since there is complete follow-up since 1977. I.e. it would be possible to follow an individual for up-to 37years.

L216-220: Or, alternatively, in some years there has been pandemics around the DST. This would not be the case if the authors had adjusted for calendar year.

L231-232: This is false. The low p-values / narrow intervals are driven by the large data set. This does not mean that the data set is free of bias, and any bias will propagate and can very well drive associations. This is ultimately the cause as to why replication and meta analysis are so vital, and it would improve the article immensely if the authors did this.

L240: Do the Swedish data set actually record the exact diagnosis date, and not just the dates of the inpatient stay?

L246: That is an interesting point. How was the sex of each patient actually identified? Could an alternative explanation not also be that some people change sex?

L252: If one is to compare the miscoding rate to the observed effects, it would be better to estimate the miscoding rate per diagnosis. Giving the overall rate does not provide a lot of information, since some diagnosis are far more frequent than others.

Reviewer #3: In this manuscript the authors demonstrate measurable health effects that are associated with the Daylight Saving Time Shift.

Using two databases, USA claims and Sweden's inpatient register, the authors found known (in literature) associations in heart diseases and injuries and also identified some novel associations related to maternal problems related to pregnancy, mental and behavioral disorders, immune-related conditions. This is a very innovative use of patient data to provide data and numbers to the debate whether DST shifts are healthy or unhealthy for populations. These associations are quite interesting and with further study can uncover additional relevant health effects associated to DST changes. Making this work highly relevant in the field.

The study design is quite clear and the methodology is exhaustively detailed within the manuscript and the abundant supplemental materials, code is available for replication. However, replicability is still not directly possible as the underlying data sources are not available (commercial and protected data) and the pre-processing of said data sources is not fully detailed or available either. The literature review is quite sufficient highlighting relevant papers to the posed problem and the methods used. While the main manuscript details are good, the real meat of this article is on the supplemental materials. The use of positive and well defined negative controls is rigorous and adequate for this type of study.

One issue that is not mentioned is that sometimes claims and EHR data is anonimzed in a way that patient timelines are date-shifted. This would greatly impact any results that have to do with temporality of patient diagnoses. While I don't believe that Marketscan is time shifted, I believe is something that should be addressed.

The signals found on the Swedish data are quite few, and it is unclear that if the relaxation they did (99.9 vs 99 CI) actually allowed them to find more (table S19 shows the Swedish signals found this way?), Finding 7 significant associations leaves plenty of the the ones found in the USA data unmatched an non evaluated(?). This makes a case to potentially use a third data source for sanity.

On potential improvement of this work is the user of a standardized common data model, such as OMOP or pcornet. This would also allow the same study to the executed at other places with data converted to the common data model and the associations could be easily validated, without the need to share any data. This would greatly enhance the impact of the contribution and provide a clear way of validating the findings.

Minor details:

Supplemental Materials (S1 Appendix) figure 1 is extremely hard to read, with the confidence intervals not clearly showing.

**Have all data underlying the figures and results presented in the manuscript been provided?**

Reviewer #1: Yes

Reviewer #2: Yes

Reviewer #3: No: Data is not available, It is understandable as one is a commercial dataset and the other is patient data. However, aggregated counts can be provided to verify the statistical claims made in the study. Confidence interval calculations are provided, but can't be verified.

PLOS authors have the option to publish the peer review history of their article (what does this mean?). If published, this will include your full peer review and any attached files.

Reviewer #1: No

Reviewer #2: No

Reviewer #3: No
---

## [Decision Letter · Decision Letter 1]

29 Apr 2020

Dear Dr. Rzhetsky,

Thank you very much for submitting your manuscript "Measurable Health Effects Associated with the Daylight Saving Time Shift" for consideration at PLOS Computational Biology. As with all papers reviewed by the journal, your manuscript was reviewed by members of the editorial board and by several independent reviewers. The reviewers appreciated the attention to an important topic. Based on the reviews, we are likely to accept this manuscript for publication, providing that you modify the manuscript according to the review recommendations.

As is evident from the reports, all reviewers where happy with the revised version of the manuscript and only two of them have minor points left. I am thus sending the manuscript back to you for minor revision, simply to give you the opportunity further strengthen the manuscript by taking this last round of input can be taken into account.

Sincerely,

Lars Juhl Jensen

Associate Editor

PLOS Computational Biology

Virginia Pitzer

Deputy Editor

PLOS Computational Biology

[LINK]

Reviewer's Responses to Questions

**Comments to the Authors:**

Reviewer #1: The authors have addressed all my mentioned concerns except one very minor issue:

They cited the prior choice recommendation in the answer to my and another reviewer. However, they chose not to include this. I would encourage the authors to cite it.

https://github.com/stan-dev/stan/wiki/Prior-Choice-Recommendations

I trust that this can be fixed without further input from my side.

Reviewer #2: I would like to commend the authors on doing a good job on clarifying uncertainties, which has led to an improvement of the manuscript. I have a few remaining comments,

The mean follow-up time for the US MarketScan database is 154, or just shy of 3 years. I think this would be valuable to add to the Methods section, as this would heavily reflect the denonimator.

Regarding the date of diagnosis, I am not sure I follow. In a 2011 description of the Swedish Inpatient Registry, the variables listed are "Admission date" and "Discharge date" (https://www.ncbi.nlm.nih.gov/pmc/articles/PMC3142234/). Typically, the date is validated on a per-disease basis, and for spinal cord injuray there was only a 62% agreement. There is no "Date of diagnosis", as far as I can see. Could the authors please clarify this? Aditionally, could the authors also comments on how an inaccuracy of the diagnosis date would reflect their results?

I am still not sure I understand the heirarchical model that the authors employ. If all RR estimates are are sampled simultaneously within the same framework, does that mean that all estimates come from a gamma distribution with mean mu? If so, is this not problematic and can lead to hyper-shrinkage? Normally this would not be an issue, but in a model as this that pools together so many different outcomes (injuries, heart disease, pregnancy), is it really expected that they all have the same mu?

I also find it odd that the authors still highlight complications in maternal care despite using a seriously flawed denominator. I think the authors would see this if they counted the number of women > 45 that are pregnant in both cohorts. If the authors are serious about this, they should validate it using a carefully crafted at-risk group.

Reviewer #3: Thanks for addressing my concerns and sharing the incidence numbers they found on the Marketscan data.

**Have all data underlying the figures and results presented in the manuscript been provided?**

Reviewer #1: Yes

Reviewer #2: Yes

Reviewer #3: Yes

PLOS authors have the option to publish the peer review history of their article (what does this mean?). If published, this will include your full peer review and any attached files.

Reviewer #1: No

Reviewer #2: No

Reviewer #3: No
---

## [Editor Report · Decision Letter 2]

6 May 2020

Dear Dr. Rzhetsky,

We are pleased to inform you that your manuscript 'Measurable Health Effects Associated with the Daylight Saving Time Shift' has been provisionally accepted for publication in PLOS Computational Biology.

Best regards,

Lars Juhl Jensen

Associate Editor

PLOS Computational Biology

Virginia Pitzer

Deputy Editor

PLOS Computational Biology

---

## [Editor Report · Acceptance letter]

1 Jun 2020

PCOMPBIOL-D-20-00022R2 

Measurable Health Effects Associated with the Daylight Saving Time Shift

Dear Dr Rzhetsky,

I am pleased to inform you that your manuscript has been formally accepted for publication in PLOS Computational Biology. Your manuscript is now with our production department and you will be notified of the publication date in due course.

With kind regards,

Laura Mallard
